# A New Method for Joint Sparse DOA Estimation

**DOI:** 10.3390/s24227216

**Published:** 2024-11-12

**Authors:** Jinyong Hou, Changlong Wang, Zixuan Zhao, Feng Zhou, Huaji Zhou

**Affiliations:** 1Key Laboratory of Electronic Information Countermeasure and Simulation Technology, Ministry of Education, Xidian University, Xi’an 710071, China; 20023110133@stu.xidian.edu.cn (J.H.); 23021211469@stu.xidian.edu.cn (Z.Z.); fzhou@mail.xidian.edu.cn (F.Z.); 2National Key Laboratory of Electromagnetic Space Security, Jiaxing 314000, China; zhouhuaji1988@sina.com

**Keywords:** DOA estimation, *l_2,th_* norm minimization, joint sparse reconstruction

## Abstract

To tackle the issue of poor accuracy in single-snapshot data processing for Direction of Arrival (DOA) estimation in passive radar systems, this paper introduces a method for judiciously leveraging multi-snapshot data. This approach effectively enhances the accuracy of DOA estimation and spatial angle resolution in passive radar systems. Additionally, in response to the non-convex nature of the mixed norm, we propose a hyperbolic tangent model as a replacement, transforming the problem into a directly solvable convex optimization problem. The rationality of this substitution is thoroughly demonstrated. Lastly, through a comparative analysis with existing discrete grid DOA estimation methods, we illustrate the superiority of the proposed approach, particularly under conditions of medium signal-to-noise ratio, varying numbers of snapshots, and close target angles. This method is less affected by the number of array elements, and is more usable in practices verified in real-world scenarios.

## 1. Introduction

The Passive Bistatic Radar (PBR) system represents a variant of traditional radar systems with historical roots dating back decades [1,2,3]. PBR possesses advantages that active radar systems lack. Firstly, it can employ “non-cooperative radiation sources” as its energy transmission source, eliminating the need for dedicated transmitters [4,5,6]. This feature yields advantages such as cost-effectiveness, stealth, resistance to interference, and extensive application prospects. Secondly, PBR can operate across a wide frequency band without interfering with existing wireless systems, utilizing external radiation sources like television, radio stations, mobile phone base stations, and digital video broadcasting [7].

Recent research in passive radar has spawned various subfields, including direct wave purification and clutter suppression. Passive radar target localization and tracking are pivotal technologies, with localization methods categorized into direction-finding [8], frequency-azimuth joint [9], and direct localization methods [10]. Many localization methods require Direction of Arrival (DOA) information, and there are numerous DOA estimation methods. This paper primarily focuses on the DOA estimation method based on compressed sensing (CS) for multiple-shot data.

Numerous researchers have proposed algorithms based on sparse models, classifying sparse representations from different perspectives [11,12,13,14,15,16,17]. For instance, methods employed in sparse constraint with different norms can be categorized into five classes: $l_{0}$ norm minimization, $l_{p}$ norm minimization $\left(0<p<1\right)$, $l_{1}$ norm minimization, $l_{2}$ norm minimization, and $l_{2,1}$ norm minimization. Furthermore, various optimization algorithms for solving the above-mentioned sparse models can also be categorized, encompassing greedy strategies, constraint optimization strategies, approximation algorithms, and homotopy algorithms.

Addressing the challenge of developing new signal processing theories for spatial spectrum estimation and spatial filtering, some scholars have analyzed the interference effects of non-ideal factors in spatial spectrum estimation from different angles. They have developed a diagonal loading algorithm, directly seeking the optimal solution for the diagonal loading value. Moreover, recent Bayesian learning-based algorithms have been proposed [18]. This method excels in accurately reconstructing signals after sparse representation in array direction pattern synthesis and angle tracking.

In [19], Bin Hu et al. proposed an offline DOA estimation method based on compressed sensing in a multipath environment. The crucial step involves transforming the signal reception model into an Errors-in-Variables (EIV) model and then estimating the sparse coefficients using total least squares. The estimated sparse coefficients are utilized to update the grid, and subsequently, the updated grid and estimated sparse coefficients are employed to estimate the multipath coefficients. In [20], Linxi Liu et al. studied the DOA estimation problem of a one-dimensional uniform linear array system with unknown inter-antenna phase errors. Leveraging CS theory, they transformed the DOA estimation problem into a sparse reconstruction problem, capitalizing on the sparsity of signals in the spatial domain. Subsequently, they proposed an iterative DOA estimation method for updating sparse signals and phase errors. In [21], Berkan Kilic et al. introduced a hardware-efficient DOA estimation method based on compressed sensing. The hardware complexity of the DOA estimation system can be tailored by design. They also considered array imperfections in numerical analysis, anticipating that the proposed design would perform effectively in practical applications. Furthermore, the performance of their proposed method can be explored in various fields of signal processing where the number of samplers is less than the number of receivers.

## 2. Main Work and Symbol Description

The paper aims to address two key problems:

Utilization of Multi-Snapshot Data in Sparse Model-Based DOA Estimation: Many existing sparse model-based Direction of Arrival (DOA) estimation methods primarily focus on single-snapshot data. However, practical applications often involve multi-snapshot data, which tends to exhibit better sparsity recovery. Therefore, the paper emphasizes the need to find a reasonable approach for the effective utilization of multi-snapshot data.

Rationalization of Sparse Recovery Models for $l_{2,1}$ and $l_{2,0}$ norms: Joint sparse recovery problems, generalizations of single measurement vector problems in compressed sensing, aim to recover sets of jointly sparse vectors. The paper explores five norm-constrained sparse reconstruction algorithms, each with distinct characteristics. Notably, due to the NP-hard nature of the $l_{2,0}$ norm model, the paper emphasizes the importance of finding a replacement, potentially by deriving a theoretical analytical expression for the $l_{2,th}$ norm minimization problem addressed herein.

The main thrust of this paper centers on the rational utilization of multi-snapshot data in passive radar engineering. In addressing the non-convexity or non-differentiability issues inherent in sparse recovery models, particularly for $l_{2,1}$ and $l_{2,0}$ norms, the paper introduces a differentiable and optimizable surrogate model, the rationality of which is rigorously verified.

Symbol definitions used in this paper include:

$∥X{∥}_{g,h}$: The g,h mixed norm of a matrix.

$X_{rowi}$: The ith row vector of a matrix.

$\langle x,y\rangle$: The inner product of two vectors.

${\left\|X\right\|}_{2,th}$: The norm model replacing the $l_{0}$ norm with the hyperbolic tangent function.

$\frac{\partial {\left\|X\right\|}_{2,th}}{\partial x_{11}(u)}$: The partial derivative of ${\left\|X\right\|}_{2,th}$ with respect to one of its variables.

## 3. Passive Radar DOA Estimation Model Based on Uniform Linear Array Antenna

In the Passive Radar System Model illustrated in Figure 1, the research scenario addressed in this paper involves a passive radar system designed to operate without emitting electromagnetic waves [22]. Instead, it harnesses the prevalent digital TV signals in space as the radiation source and employs a Uniform Linear Array (ULA) as the receiving station. The electromagnetic waves in space encompass not only target-reflected echoes, but also direct signals and multipath clutter, further compounded by noise within the receiving array. Given that most passive radar systems initially undertake clutter suppression and direct signal cancellation, the signal model proposed in this paper assumes a scenario where, post-elimination of multipath clutter and direct signals from the receiving array, only the target echoes and residual noise persist.

In Figure 2, a one-dimensional Uniform Linear Array (ULA) is depicted, comprising M amplitude and phase-consistent elements selected as the reference element, with an element spacing of d. Considering K far-field narrowband signals incident on the array from various directions, the signals received by the mth elements and the kth signals at time t are expressed as [23]:(1)$$y_{m}=\sum_{k=0}^{K-1}S_{k}\left(t-\tau_{m,k}\right)+n_{m}\left(t\right)\;\;\;\;m=1,2,\ldots ,M$$

Here, $\tau_{m,k}$ denotes the time delay of the signal $S_{k}\left(t\right)$ when it reaches the mth element relative to the main element. For the ULA, the delay is given by $\tau_{m,k}=\left(m-1\right)d\sin\theta_{k}/c$, where c represents the speed of electromagnetic waves in a vacuum, generally considered as $3\times 10^{8},m/s$. The term $n_{m}\left(t\right)$ signifies the noise received by the mth element or generated by the array itself, with the assumption that all noises are independent Gaussian white noises.

The expressions for receiving signals at each element are:(2)$$y_{m}=\overset{K-1}{\underset{k=0}{\sum }}e^{-j2\pi f_{0}\tau_{m,k}}S_{k}+n_{m}\left(t\right)\;\;\;\;m=1,2,\ldots ,M$$
where $f_{0}$ is the carrier frequency of the signal, $j=\sqrt{-1}$.

Writing the above equation in matrix form, we obtain the signals received by the uniform array at time [24]:(3)$$y\left(t\right)={\left[y_{1}\left(t\right),y_{2}\left(t\right),\ldots ,y_{M}\left(t\right)\right]}^{T}$$
(4)$$\begin{array}{c} y\left(t\right)=\left[\begin{array}{cccc} 1 & 1 & \ldots & 1\\ e^{-j(2\pi d\sin\theta_{1}/\lambda )} & e^{-j(2\pi d\sin\theta_{2}/\lambda )} & \ldots & e^{-j(2\pi d\sin\theta_{K}/\lambda )}\\ \ldots & \ldots & \ldots & \ldots \\ e^{-j(2\pi \left(M-1\right)d\sin\theta_{1}/\lambda )} & e^{-j(2\pi \left(M-1\right)d\sin\theta_{2}/\lambda )} & \ldots & e^{-j(2\pi \left(M-1\right)d\sin\theta_{K}/\lambda )} \end{array}\right]\\ \;\;\;\;\;\;\;\;\;*\left[\begin{array}{c} S_{1}\left(t\right)\\ S_{2}\left(t\right)\\ \ldots \\ S_{K}\left(t\right) \end{array}\right]+\left[\begin{array}{c} n_{1}\left(t\right)\\ n_{2}\left(t\right)\\ \ldots \\ n_{K}\left(t\right) \end{array}\right] \end{array}$$

The above equation can be expressed as:(5)$$y\left(t\right)=A\left(\theta \right)s\left(t\right)+n\left(t\right)$$
where
(6)$$s\left(t\right)={\left[S_{1}\left(t\right),S_{2}\left(t\right),\ldots ,S_{K}\left(t\right)\right]}^{T}$$
is the incident signal,
(7)$$n\left(t\right)={\left[n_{1}\left(t\right),n_{2}\left(t\right),\ldots ,n_{K}\left(t\right)\right]}^{T}$$
is the noise signal,
(8)$$A\left(\theta \right)={\left[a\left(\theta_{1}\right),a\left(\theta_{2}\right),\ldots ,a\left(\theta_{K}\right)\right]}^{T}$$
is the flow matrix of the array, which is determined by the geometric structure of the array and the incident angle together, and
(9)$$a\left(\theta \right)={\left[1,e^{-j(2\pi d\sin\theta /\lambda )},\ldots ,e^{-j(2\pi d\sin\theta /\lambda )}\right]}^{T}$$
is the steering vector of the array.

Practical array signal processing is basically a multi-snapshot situation, and the expression of the array received signal can be written as follows:(10)$$Y=A\left(\theta \right)S+N$$
where
(11)$$Y={\left[y\left(t_{1}\right),y\left(t_{2}\right),\ldots ,y\left(t_{T}\right)\right]}^{T}$$
is the array output signal,
(12)$$S={\left[s\left(t_{1}\right),s\left(t_{2}\right),\ldots ,s\left(t_{T}\right)\right]}^{T}$$
is the incident signal matrix, and
(13)$$N={\left[n\left(t_{1}\right),n\left(t_{2}\right),\ldots ,n\left(t_{T}\right)\right]}^{T}$$
is the noise signal matrix. T is snapshots.

There, A∈ℂM×N is the observation matrix K<M≪N, that is, the sparsity of the signal to be recovered is much less than the size of the observation matrix, so the final received signal can be linearly represented by the following equation:(14)Y=y1,…,yk=A(θ)X

This is a norm-based solution method. When some measurement vectors share the same support set, we will recover a joint sparse signal from multiple measurement vectors.
(15)$$\min∥X{∥}_{g,h}\mathrm{subject}\;\mathrm{to}Y=AX$$

We can obtain (15), where $∥X{∥}_{g,h}$ is defined as follows:(16)$$∥X{∥}_{g,h}={\left[\sum_{\tilde{j}=1}^{N}{\left(\sum_{\tilde{i}=1}^{k}{\left|X\left[\tilde{j},\tilde{i}\right]\right|}^{g}\right)}^{h/g}\right]}^{1/h}$$

When g=2 and h=0 are equal, it is the $l_{2,0}$ norm, and when g=2 and h=1 are equal, it is the $l_{2,1}$ norm. Here arises a question, whether multi-snapshot data can use the idea of column blocking by restoring each shot’s original signal to calculate the joint sparse signal. The following example can fully negate this idea.

Assuming the observation matrix is
(17)$$A=\left[\begin{array}{ccccc} 2 & 0 & 0 & 1 & 0\\ 0 & 0.5 & 0 & 1 & 0\\ 0 & 0 & 1 & 2 & -0.5\\ 0 & 0 & 0 & -1 & 0.5 \end{array}\right]$$
and the signal received by the signal processor in the DOA estimation problem is
(18)$$B={\left[\begin{array}{c} 1\;\;1\;\;0\;\;0\;\;\\ 1\;\;1\;\;1\;\;0\;\; \end{array}\right]}^{T}$$

The observation matrix partitions the original signal into spatial angle divisions, ranging from four to five divisions. This division process facilitates the recovery of the initial input signal X sparsely through the equation AX=B. The matrix B is subsequently column-blocked into two column vectors, denoted as $B=[b_{1},b_{2}]$.

To elaborate, the individual components of B are organized into two distinct vectors. Specifically, the two single-shot input signals, represented by $x_{1}={\left[0.5\;2\;0\;0\;0\right]}^{T}$ and $x_{2}={\left[0\;0\;1\;2\;0\right]}^{T}$, are determined by utilizing a sparse recovery method applied to the respective equations $Ax_{1}=b_{1}$ and $Ax_{2}=b_{2}$.

Subsequently, the $l_{2,0}$ norm of the concatenated vector $\left[x_{1},x_{2}\right]$ is computed, resulting in a value of 4 (indicating the number of non-zero row vectors after the matrix is block diagonalized). However, it is noteworthy that there exists a more parsimonious solution with fewer sparse degrees. By denoting the five angles in space as [1°,2°,3°,4°,5°], the estimated Direction of Arrival (DOA) of the target obtained through two single-shot data processing methods is denoted as single-shot [1°,2°,3°,4°] and [1°,2°,3°], while the estimated DOA obtained through joint sparse recovery is represented as joint recovery.

Upon analysis, it is evident that joint recovery provides a more accurate estimation compared to single-shot [25]. Consequently, this example highlights that relying solely on multiple single-shot data processing methods may not guarantee the desired solution. The specific issue demonstrated in this example could potentially arise in real Direction of Arrival (DOA) estimation problems. Hence, determining effective strategies for leveraging multiple fast-shot data becomes a crucial and nuanced aspect of addressing such challenges.

## 4. Proof of Model Equivalence and Proposed Algorithm

It can be seen that an underdetermined equation AX=B has infinite solutions, but for DOA estimation in this paper, the sparsest solution is always wanted, and there are many methods to find this sparsest solution. This paper mainly studies the sparse solution of the equation under mixed norm constraints.

Since the five classifications based on the minimum norm model analyzed before each have advantages and disadvantages, the l2,th norm model is analyzed here. Inspired by l0 minimization and lp minimization model transformation, this is a method for finding optimal solutions to MMV problems, which solves the following minimum optimization problems
(19)minX∈ℝn×r∥X∥2,th subject to AX=B

The $l_{0}$ minimization problem is a problem that has been shown to be NP-hard due to discreteness, so the $l_{2,0}$ minimization problem is also NP-hard. However, $∥X{∥}_{2,0}=\lim_{\alpha \to \infty }∥X{∥}_{2,th_{\alpha }}$, the $l_{2,th}$ minimization problem is theoretically solvable, and the proof of this problem has been reflected in the literature [26]. The question of whether the solutions obtained by $l_{2,th}$ minimizing and $l_{2,0}$ minimizing the two models are equivalent will be discussed next.

The $l_{0}$ norm of a given vector is defined as
(20)$${\left\|x\right\|}_{0}=\left\{\begin{array}{l} 1\langle x,x\rangle \neq 0\\ 0\langle x,x\rangle =0 \end{array}\right.$$

Obviously, this piecewise function is not differentiable, so this paper approximates the $l_{0}$ norm with the following functions, where α>0.
(21)$${\left\|x\right\|}_{0}\approx \sum_{i=1}^{n}\tanh\alpha x_{i}=\sum_{i=1}^{n}\frac{e^{\alpha x_{i}}-e^{-\alpha x_{i}}}{e^{\alpha x_{i}}+e^{-\alpha x_{i}}}$$

The graph of the hyperbolic tangent function is shown in the Figure 3, and it can be seen that the larger the value α, the closer the function of the $l_{0}$ norm.

The $l_{2}$ norm of a vector is defined as
(22)$${\left\|x\right\|}_{2}=\sqrt{\sum_{i=1}^{n}{\left(x_{i}\right)}^{2}}$$

The mixed $l_{2,0}$ norm of a matrix is defined as
(23)$${\left\|X\right\|}_{2,0}={\left\|\left[{\left\|X_{row1}\right\|}_{2};{\left\|X_{row2}\right\|}_{2}\ldots ;{\left\|X_{rowm}\right\|}_{2}\right]\right\|}_{0}$$

Bring in the $l_{0}$ norm model approximated by the hyperbolic tangent function and define it as
(24)$${\left\|X\right\|}_{2,0}\approx \sum_{i=1}^{m}\tanh\alpha {\left\|x_{rowi}\right\|}_{2}={\left\|X\right\|}_{2,th}$$

Some ${\left\|X\right\|}_{2,th}$ properties discussed below illustrate their equivalence with the $l_{2,0}$ norm model.

**Theorem** **1.**
*For a given matrix X∈ℝm×n, X2,th there is an upper bound M and satisfied M≤m, there is a lower bound 0.*


A lemma needs to be proven before we can prove this theorem.

**Lemma** **1.**
*$f(\alpha x)=\frac{e^{\alpha x_{i}}-e^{-\alpha x_{i}}}{e^{\alpha x_{i}}+e^{-\alpha x_{i}}}\;,\alpha >0$ is a monotonically increasing function under x∈[0,+∞] and its value is given by [0,1).*


Without loss of generality, consider a function $f(x)=\frac{e^{x}-e^{-x}}{e^{x}+e^{-x}}$ of one variable, where [−∞,+∞] is a derivable function, whose derivative is,
(25)$$f^{\prime }(x)=1-f^{2}(x)$$

Since $\underset{x\to +\infty }{\lim}f(x)=1$ and $\max\left|f(x)\right|=1$, it follows that $f^{\prime }(x)\geq 0$; therefore, f(x) is a monotone nondecreasing function with upper bound 1, and according to $\underset{x\to 0}{\lim}f(x)=0$, 0≤f(x)≤1x∈[0,+∞].

And then according to ${\left\|X\right\|}_{2,th}=\sum_{i=1}^{m}\tanh{\left\|x_{rowi}\right\|}_{2}$, considering ${\left\|x_{rowi}\right\|}_{2}$ is the $l_{2}$ norm of the row vector of a matrix X, and by the definition of $l_{2}$ norm, therefore ${\left\|x_{rowi}\right\|}_{2}$ is a bounded nonnegative number. And then according to the property of f(x), for any $x_{rowi}$, the following equations are
(26)$$0\leq \tanh{\left\|x_{rowi}\right\|}_{2}\leq 1$$

Since X∈ℝm×n, when all xrowi2→+∞, X2,th→m, when all xrowi2=0, X2,th=0. There is an X2,th upper bound M and satisfies M≤m, and there is a lower bound 0. Proof done.

**Theorem** **2.**
*For a given matrix X∈ℝm×n, X2,th is a continuously derivable function for every element in the matrix.*


To prove Theorem 2, we need to assume a time-varying matrix and, without loss of generality, let the matrix have only one variable, so we can assume that the matrix is as follows,
(27)$$X=\left[\begin{array}{cccc} x_{11}(u) & x_{12} & \ldots & x_{1n}\\ x_{21} & \ldots & \ldots & \ldots \\ \ldots & \ldots & \ldots & \ldots \\ x_{m1} & \ldots & \ldots & x_{mn} \end{array}\right]$$

And then compute it ${\left\|X\right\|}_{2,th}$,
(28)$${\left\|X\right\|}_{2,th}=\tanh(\sqrt{{\left(x_{11}(u)\right)}^{2}+\sum_{i=2}^{n}x_{1i}^{2}})+\sum_{i=2}^{m}\tanh(\sqrt{\sum_{k=1}^{n}x_{ik}^{2}})$$

Since ${\left(•\right)}^{n}$ and tanh(•) are both continuous functions, according to Lemma 2 below,

**Lemma** **2.**
*A composition of continuous functions is a continuous function in its domain.*


Therefore, ${\left\|X\right\|}_{2,th}$ is a continuous function, and according to the derivative formula of the composite function, its derivative is
(29)$$\frac{\partial {\left\|X\right\|}_{2,th}}{\partial x_{11}(u)}=\left[1-\tanh^{2}(\sqrt{{\left(x_{11}(u)\right)}^{2}+\sum_{i=2}^{n}x_{1i}^{2}})\right]\left[\frac{x_{11}(u)}{\sqrt{{\left(x_{11}(u)\right)}^{2}+\sum_{i=2}^{n}x_{1i}^{2}}}\right]$$

Proof is completed.

**Theorem** **3.**
*For a given matrix X∈ℝm×n, when X2,th xij∈[0,+∞) is a monotone nondecreasing function for every element of the matrix.*


To prove that Theorem 3 holds, we need to show that the ${\left\|X\right\|}_{2,th}$ derivative of Equation (29) exists $\frac{\partial {\left\|X\right\|}_{2,th}}{\partial x_{11}(u)}\geq 0$ for any $x_{11}(u)\in [0,+\infty )$. Analyze the second term in Equation (29), when $x_{11}(u)\in (0,+\infty )$ $\sqrt{{\left(x_{11}(u)\right)}^{2}+\sum_{i=2}^{n}x_{1i}^{2}}>0$, and when $x_{11}(u)=0$ $\frac{x_{11}(u)}{\sqrt{{\left(x_{11}(u)\right)}^{2}+\sum_{i=2}^{n}x_{1i}^{2}}}=0$, analyze the first term in Equation (29) by the properties of Equation (25) in Lemma 1; we know that when $x_{11}(u)\in [0,+\infty )$, $1-\tanh^{2}\left(\sqrt{{\left(x_{11}(u)\right)}^{2}+\sum_{i=2}^{n}x_{1i}^{2}}\right)\geq 0$.

To sum up, when ∂X2,th∂x11(u)≥0 holds if x11(u)∈[0,+∞) and only if x11(u)=0, the equality sign holds. So, for the given matrix X∈ℝm×n, when X2,th xij∈[0,+∞), it is a monotone nondecreasing function for every element of the matrix. Proof done.

According to the proofs of Theorems 1–3, it is obvious that the $l_{2,th}$ model proposed in this paper is equivalent to the $l_{2,0}$ model.

Now, in order to design an algorithm to estimate the joint sparse matrix X, the following two theorems are introduced and proven to show the rationality of the sparse reconstruction of the matrix by solving the $l_{2,th}$ norm minimization.

**Theorem** **4.***For the matrix Y∈ℝm×p under the constraint Y=AX, it satisfies:*(30)X=ΛXATAΛXAT†Y*where* ΛX∈ℝm×m *is a diagonal matrix whose entries on the diagonal *ΛXi,i *are:*(31)ΛXi,i=Xrowi2tanh′Xrowi2

The proof of Theorem 4 reconsiders the following questions:(32)$$\underset{X}{\min}∥X{∥}_{2,th}\mathrm{subject}\;\mathrm{to}\;Y=AX$$

$∥X{∥}_{2,th}$ derivability with respect to each variable has already been proven in Theorem 2, so when using the KKT condition on Equation (32), a Lagrange function is defined here:(33)$$L\left(X,\lambda \right)=∥X{∥}_{2,th}-\lambda \left(AX-Y\right)$$

It can be seen that the estimation algorithm is a form of fixed point iteration, so it is necessary to discuss the convergence of the algorithm. Namely, Theorem 5 below.

**Theorem** **5.**
*For a sequence of matrices $\left\{X_{t}\right\}$ satisfying the following recurrence relation:*

(34)
$$X_{t+1}=\Lambda \left(X\right)A^{T}{\left(A\Lambda \left(X_{t}\right)A^{T}\right)}^{†}Y$$



Satisfies the following inequality:(35)$$∥X_{t+1}{∥}_{2,th}\leq ∥X_{t}{∥}_{2,th}$$

And $\underset{t\to \infty }{\lim}X_{t}=X_{end}$; this limit value is the final estimated value of Equation (34).

Now, Theorem 5 is proven. According to the Formula (30), the following conditions can be obtained:(36)$$\left\{\begin{array}{l} \Lambda^{†}\left(X\right)X_{t+1}=A^{T}{\left(A\Lambda \left(X_{t}\right)A^{T}\right)}^{†}Y\\ Y=AX_{t+1} \end{array}\right.$$

The Formula (36) can be transformed into the following minimum quadratic optimization problem:(37)$$\min X^{T}\Lambda^{†}\left(X_{t}\right)X\;\mathrm{subject}\;\mathrm{to}\;Y=AX$$

Namely,
(38)$$\sum_{i=1}^{m}\frac{\tanh\left({\left\|X_{t,rowi}\right\|}_{2}\right)}{{\left\|X_{t,rowi}\right\|}_{2}}{\left({\left\|X_{t+1,rowi}\right\|}_{2}\right)}^{2}\leq \sum_{i=1}^{m}\tanh\left({\left\|X_{t,rowi}\right\|}_{2}\right){\left\|X_{t,rowi}\right\|}_{2}$$

From Theorem 3 obtained in the previous proof, we know that ${\left\|X\right\|}_{2,th}$ in $x_{ij}\in [0,+\infty )$ is a monotone nondecreasing function, so we can obtain:(39)$$\begin{array}{l} \tanh\left({\left\|X_{t+1,rowi}\right\|}_{2}\right)-\frac{{\left\|X_{t+1,rowi}\right\|}_{2}^{2}\tan h^{\prime }\left(X_{t}\right)}{2{\left\|X_{t,rowi}\right\|}_{2}}\\ \leq \tanh\left({\left\|X_{t,rowi}\right\|}_{2}\right)-\frac{{\left\|X_{t,rowi}\right\|}_{2}\tan h^{\prime }\left(X_{t}\right)}{2} \end{array}$$

According to Theorem 1, the iteration result has a lower bound 0, so the Formula (38) is proven and Theorem 5 holds. In this paper, a DOA estimation method based on l2,th minimization is proposed as Algorithm 1.
**Algorithm 1.** Second mixed norm constraintInput: observed signal Y, measurement matrix A, signal sparsity K.Output: An estimate of the signal X*.Initialization: Number of iterations t=1, let M1=In.In the tth iteration: Step 1: Update the estimate of the original signal Xt+1*=Mt−1AT(AMt−1AT)−1Y.Step 2: Use the Xrowi2tanh′Xrowi2 i=1,2,…,n updated Mt+1−1 main diagonal elements Step 3: Stop iterating on number of t>K iterations or convergence 

## 5. Numerical Experiment

The radiation source employed in this study is based on the digital TV signal DVB-T standard, characterized by a carrier frequency of 522 MHz and a signal bandwidth of 100 kHz. Figure 4 depicts a simulation of the corresponding baseband signal.

Furthermore, in Figure 5, the antenna pattern function of a single array element is illustrated, showcasing a consistent gain across the angular space of [1°,90°]. Where the blue line is the normalized gain of the antenna on all directions. This observation emphasizes the uniformity in gain within this specific angle range.

The experimental scenario in the simulation experiment unfolds in a two-dimensional plane space spanning an area of $2500\times 1000\;{\mathrm{km}}^{2}$. The simulation parameters are delineated as Table 1:

In the first simulation experiment as Figure 6, the spatial angle is discretized into 891 grid points within the range of [1°,90°]. The anticipated location of the target is set at 30°. The experimental findings reveal that, under a Signal-to-Noise Ratio (SNR) of 10 dB, all three Direction of Arrival (DOA) estimation algorithms consistently and accurately estimate the DOA of the target.

In the second simulation experiment as Figure 7, the number of targets in the spatial domain was increased, and the assumed Directions of Arrival (DOA) for the target were set at [10°,20°,30°]. They are displayed as the three colored circles in the figure. All other conditions remaining are unchanged. The obtained result figures indicate that all three methods accurately estimate the DOA of the targets.

In the actual passive radar DOA estimation, due to the limitation of hardware conditions, it is often impossible to have a lot of array elements, so the performance of the proposed DOA estimation method is particularly important when the array elements are insufficient.

In the simulation experiment in Figure 8, the number of elements is set to 8, and it can be seen that the $l_{2,0}$ M-OMP method cannot correctly estimate the target DOA at all, and the $l_{2,1}$ base-matching tracking method can estimate the target DOA, but the main lobe of the spectrum is too wide, while the method proposed $l_{2,th}$ in this paper can accurately estimate the target DOA.

In the simulation experiment in Figure 9, the number of elements is further reduced, and the number of elements is set to 4. It can be seen that the M-OMP method still fails to correctly estimate the target DOA. Although the base matching tracking method can estimate the target DOA, the error of DOA estimation will be large as the main lobe of the spectrum becomes wider. However, the proposed method can still accurately estimate the target DOA.

DOA estimation methods based on sparse recovery generally require prior information about the number of targets. However, in the actual passive radar system, this prior information cannot be obtained most of the time, and the sparse recovery is generally carried out using a fixed number of iterations. Therefore, whether the algorithm can stably estimate the DOA of the target with the increase in the number of iterations is particularly important.

Figure 10 shows the result of setting the number of iterations to 60. It can be seen that the M-OMP method cannot correctly estimate the target DOA at all, while the base-matching tracking method can estimate the target DOA but has some secondary peaks, while the method proposed in this paper can accurately estimate the target DOA.

We want to apply the algorithms proposed in the theory to practical problems, so it is necessary to conduct simulation experiments that approximate the real environment. We simulate a real passive radar scenario by setting the number of array elements to 8, the number of sources to 1, and the signal-to-noise ratio to 0 dB. Considering that phase error often exists in real equipped receiving systems, this phase error may be due to the spacing of antenna array elements presenting an unknown non-uniformity in the production or use, or it may also be caused by the amplitude of different channels of the hardware for the conversion of analog to digital signals in the receivers; phase inconsistencies result. Therefore, the next simulation experiments consider the effect of random phase errors with a 60° Gaussian distribution for each array element on the DOA estimation.

Figure 11 shows the simulation results of the proposed method and the two methods compared in the relatively realistic scenario in the above. We set the iteration termination condition to 15 times and all three algorithms have outputs at the real DOA position of the target. The M-OMP results have many false peaks with an amplitude of more than 0.5, which are difficult to identify due to the high accuracy requirements of the M-OMP method on the antenna and receiver. The basis pursuit results have relatively few false peaks with an amplitude of more than 0.5, but have too many burrs, which is not conducive to the actual DOA estimation. The proposed method can accurately estimate the DOA, but there are fluctuations around the true DOA of the target, which is generally better than the two compared methods and is more likely to be applied in practice.

This study conducts a comparative analysis of the root mean square error (RMSE) in Angle estimation for three sparse recovery algorithms. The evaluation is performed under various signal-to-noise ratios and differing sparsity and snapshot numbers. The RMSE is defined as follows:(40)$$E_{RMS}=\left(\frac{1}{M}\sqrt{\sum_{i=1}^{M}{\left(\overset{\wedge }{\theta }-\theta_{real}\right)}^{2}}\right)$$

In the context of the following experiments, $\overset{\wedge }{\theta }$ represents the outcome of Direction of Arrival (DOA) estimation, $\theta_{real}$ signifies the true angle of arrival, and M stands for the number of Monte Carlo experiments, consistently set to 100 in all subsequent trials.

In the first comparative experiment as Figure 12, the RMSE curves for the Direction of Arrival (DOA) estimation of three targets are presented over a signal-to-noise ratio range of [−30 dB,0 dB]. The overall RMSE of the three methods shows a decreasing trend with the signal-to-noise ratio. The accuracy of the M-OMP method improves rapidly as the signal-to-noise ratio increases. The basis pursuit method improves accuracy slowly. The figure in this paper illustrates that, as the Signal-to-Noise Ratio (SNR) increases, the mean square error of the proposed second-norm constraint algorithm surpasses that of the $l_{2,0}$ and $l_{2,1}$ minimization methods, particularly in low SNR scenarios. However, in high SNR situations, the performance of the proposed algorithm does not outperform that of the M-OMP algorithm. This can be attributed to the fact that the norm model is a discontinuous function and experiences rapid changes in ideal conditions.

In the second comparative experiment as Figure 13, we investigated the impact of the angular separation between two targets on the measurement error. Specifically, the reference angle of one target is fixed at 10°, while the angle of the other target is set as (10+Δθ)°. Simulation results reveal that the proposed algorithm exhibits superior resolution and yields the smallest Root Mean Square Error (RMSE) for the target positioned closer in angular space. It is worth mentioning that the basis pursuit method performs poorly when the target angle difference is small, but performs well when the target angle difference is large, and is suitable for DOA estimation when the targets are far apart in azimuthal angle.

In the third comparison experiment as Figure 14, we set up a change in the number of snapshots. Although the three DOA estimation methods are based on different algorithmic implementations, they may all be subject to the same constraints when these methods are compared at different numbers of snapshots. Under these constraints, the performance differences between the methods may be attenuated, leading to convergence of results. For example, if all the methods are constrained by the array element or the signal-to-noise ratio, their errors may fluctuate within a similar range. However, the proposed method fluctuates less and is relatively stable compared to the basis pursuit method. The good performance of the proposed method compared to the two methods compared at low snapshots is due to the nonlinear convex function.

In the fourth comparison experiment as Figure 15, we set up different numbers of array elements to analyze the RMSE of the three methods. Other conditions remain unchanged; the goniometric error usually tends to decrease with the increase in the number of array elements because more array elements can provide better spatial resolution and thus improve the accuracy of the measurement. Significant differences in goniometric errors may exist between different methods. M-OMP is more sensitive to the number of array elements and has a large error when the number of array elements is small, which decreases significantly after the number of array elements increases. This is related to its non-convexity. The basis pursuit method and the proposed method are not sensitive to the number of array elements, and overall, the proposed method performs better than the basis pursuit method, and the error of the proposed method decreases significantly when the number of array elements is higher than 12, which indicates that it is more usable in the actual low number of array element devices. This indicates that it is more usable in real devices with a low number of array elements. Different methods or algorithms may have significant differences in the goniometric errors under the same number of array elements, and the optimal method or algorithm should be selected according to the actual application scenarios and requirements.

As shown in Figure 16a, the antenna used in this field experiment is an eight-array antenna composed of two four-array antennas, with the main flap width recorded as [1°,90°], placed on the roof of the Hangzhou Research Institute of Xidian University, with the normal direction recorded as 45°, and the three locations of the Beigaofeng TV tower in Hangzhou, Hangzhou Research Institute of Xidian University, and Hangzhou Xiaoshan Airport are calibrated, as shown in Figure 16b. We set the normal direction of the antenna to the east direction (the direction of the black line in the figure) and use the 530 MHZ DVB-T signal transmitted from Hangzhou Beifeng TV tower to estimate the DOA of the echoes of the civil aviation targets taking off and landing at Hangzhou Xiaoshan Airport, under which the target can be regarded as a point target. As shown in Figure 17, the azimuth angle of the target that should appear according to the geographic location is 48°, while the azimuth angle measured by using the method proposed in this paper is 47.3°, which is 0.7° different from the real position.

## 6. Conclusions

To facilitate the judicious application of multi-snapshot data in the context of Direction of Arrival (DOA) estimation in passive radar, this paper undertakes a comprehensive exploration. Initially, it examines the correctness and efficacy of multi-snapshot data in contrast to single-snapshot data. Subsequently, acknowledging the non-convex nature of the norm model, an alternative model is introduced. The correctness and rationality of this alternative model are demonstrated, effectively addressing the NP-hard problem associated with norm optimization.

In conclusion, the proposed method’s rationality is substantiated through simulation experiments. Comparative analysis with classical sparse methods reveals that, under specific signal-to-noise ratio conditions, the proposed method exhibits smaller errors, is less sensitive to the number of snapshots, and demonstrates enhanced resolution in the DOA estimation of targets with similar spatial angles. The robustness of the proposed method in this paper is greatly improved compared to M-OMP and is more practical in real-life scenarios. We have completed experiments in real-life scenarios and achieved certain results.

In this research article, we designed a new algorithm for DOA estimation based on the proposed new model. Currently, within the more ideal case analysis, we believe that there is still a lot of room for improvement in the method. The current work is only a phase, and we will conduct more experiments in real environments so that the method can really be applied in practice.

Although this study tries to be rigorous and meticulous, due to the limitations of time, resources, and personal knowledge, there are inevitably omissions and shortcomings. It is hoped that this study will have the effect of drawing attention to these shortcomings.

## Figures and Tables

**Figure 1 sensors-24-07216-f001:**
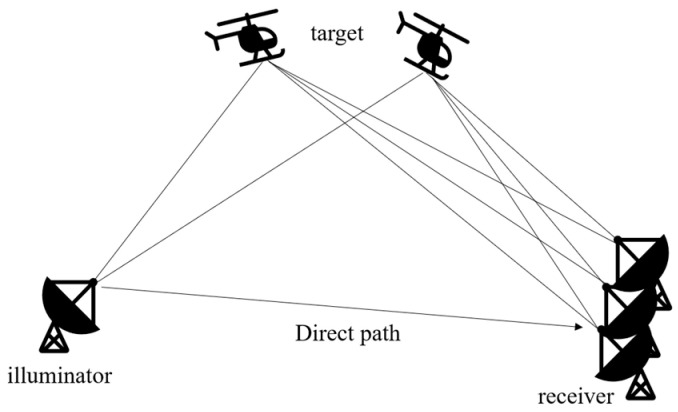
Passive radar system model.

**Figure 2 sensors-24-07216-f002:**
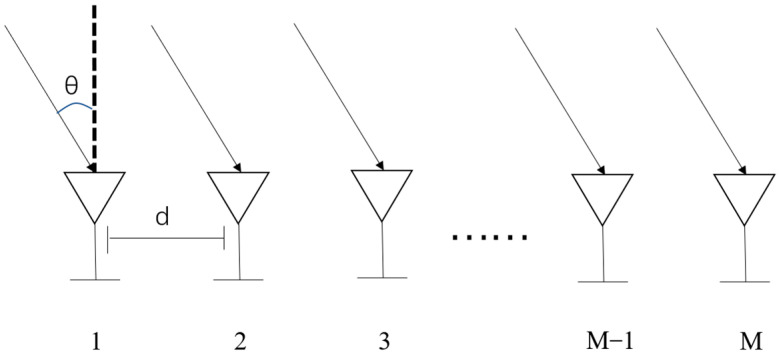
Uniform array antenna model.

**Figure 3 sensors-24-07216-f003:**
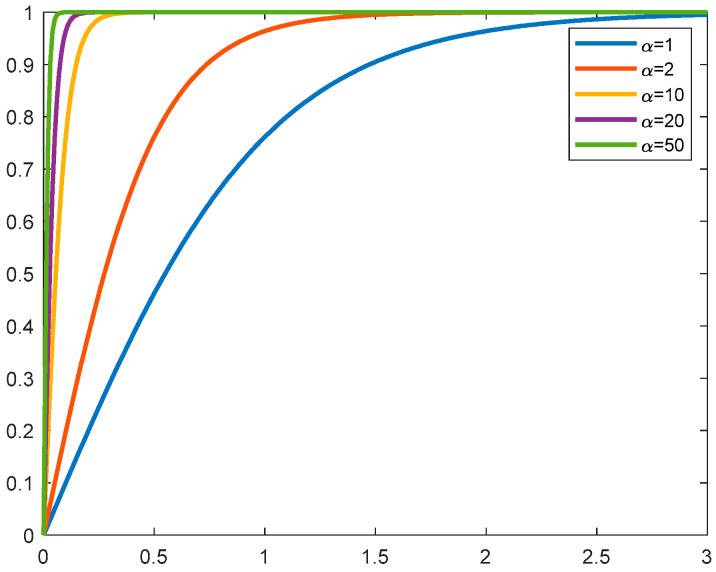
Hyperbolic tangent functions with different parameters.

**Figure 4 sensors-24-07216-f004:**
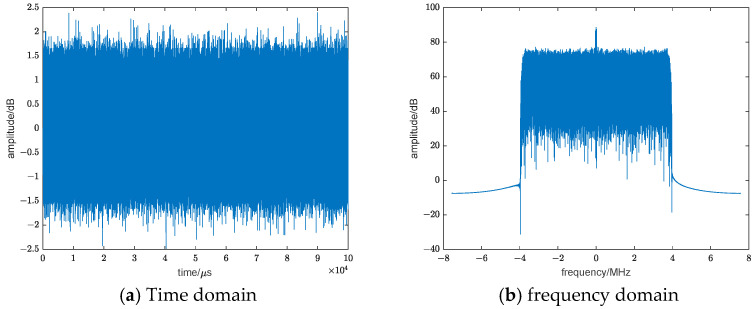
Digital TV signal simulation.

**Figure 5 sensors-24-07216-f005:**
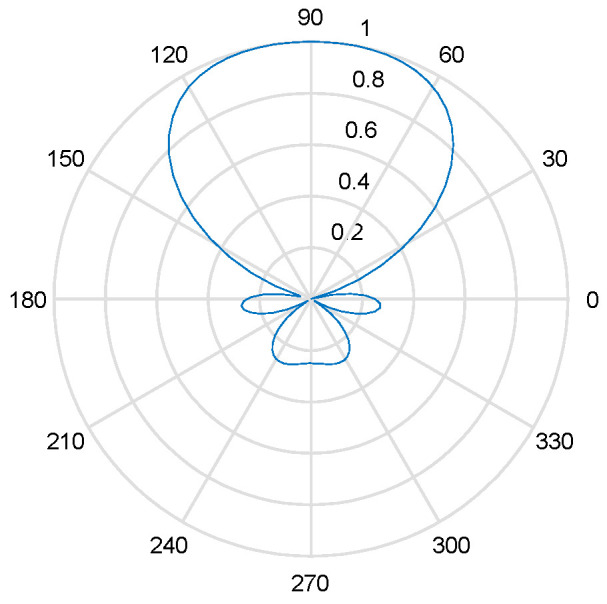
Single array antenna pattern.

**Figure 6 sensors-24-07216-f006:**
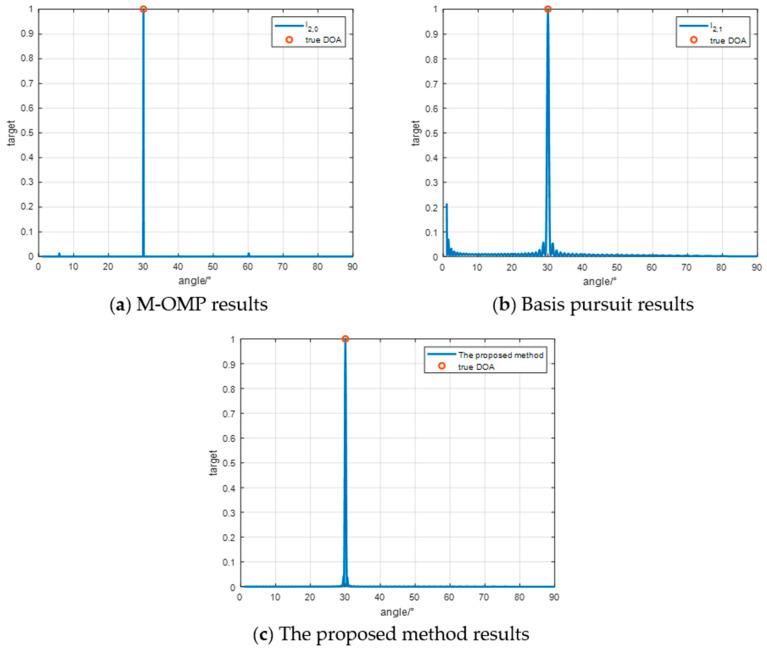
Single target effect experiment.

**Figure 7 sensors-24-07216-f007:**
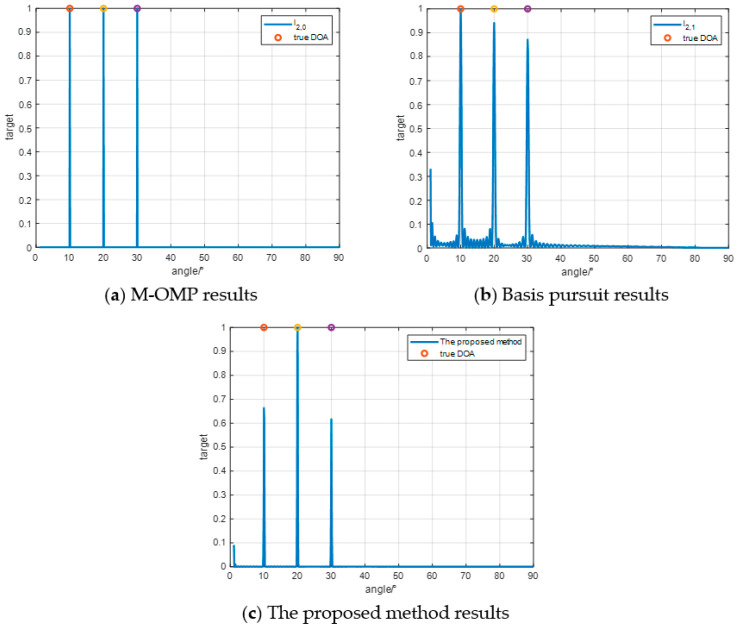
Three-objective effect experiment.

**Figure 8 sensors-24-07216-f008:**
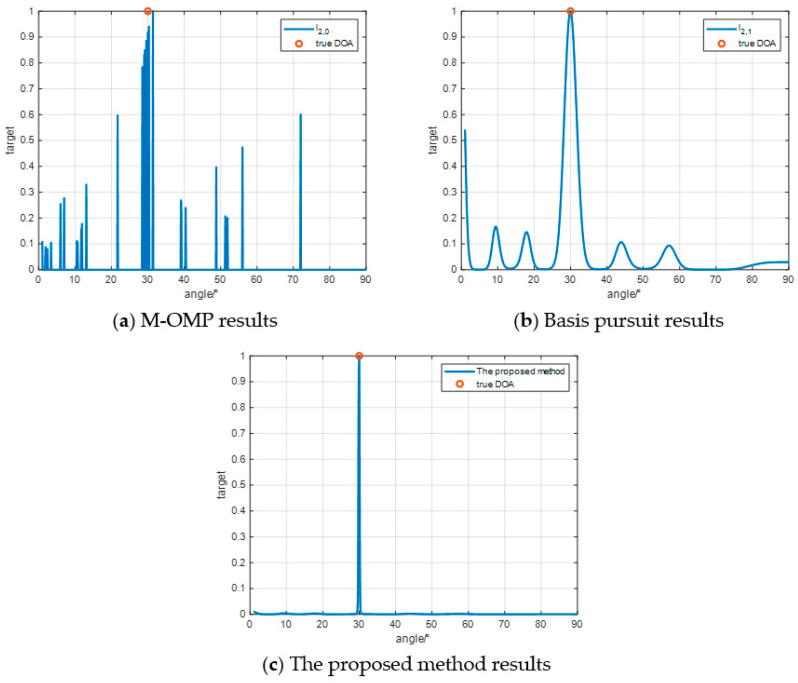
Eight matrix DOA estimation results.

**Figure 9 sensors-24-07216-f009:**
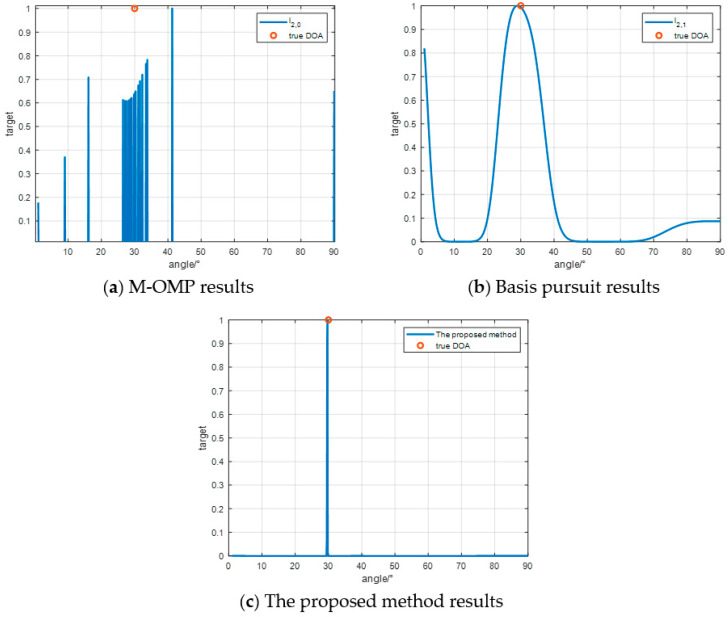
DOA estimation results of four array elements.

**Figure 10 sensors-24-07216-f010:**
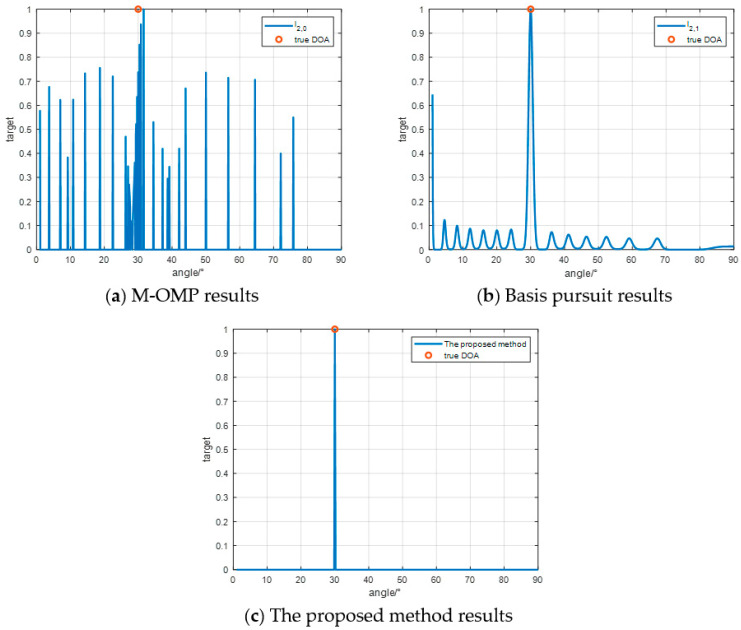
Results of 60 iterations.

**Figure 11 sensors-24-07216-f011:**
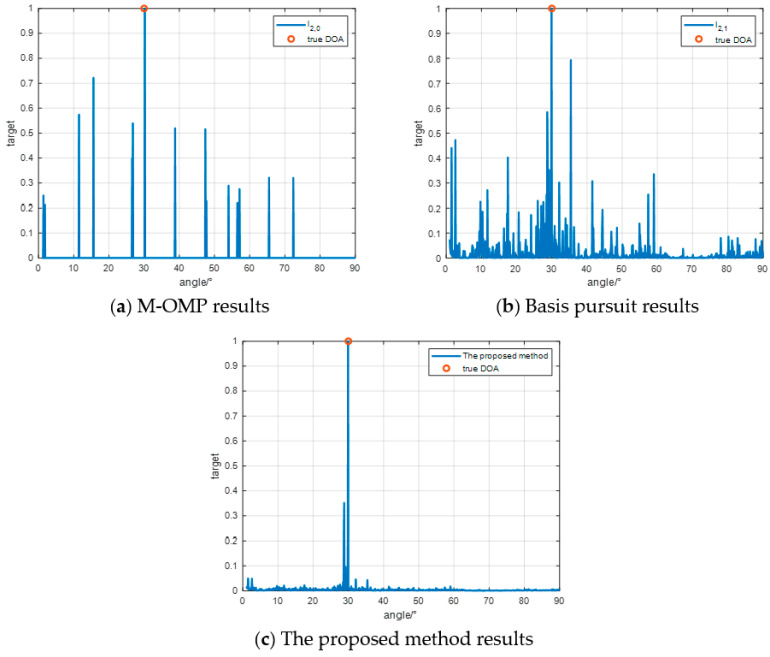
Simulation analysis similar to the real environment.

**Figure 12 sensors-24-07216-f012:**
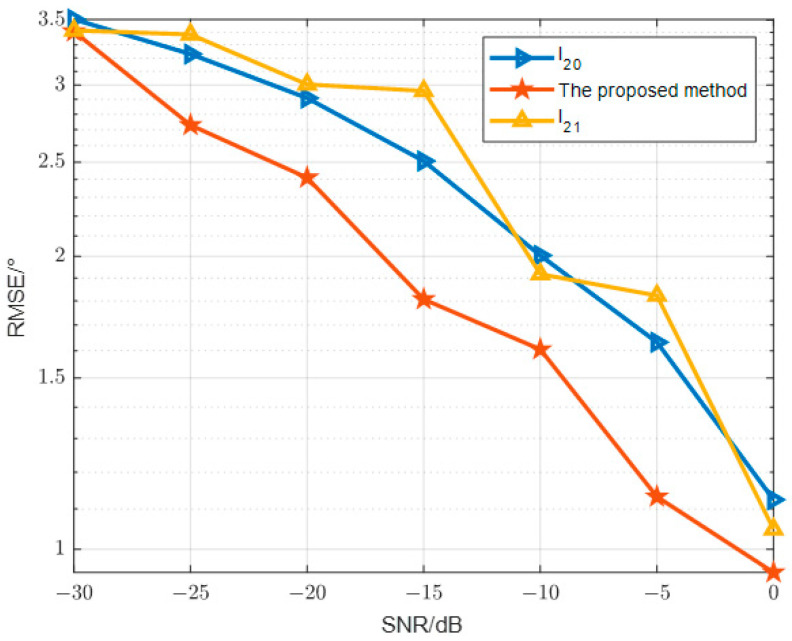
Relationship between direction finding error and SNR.

**Figure 13 sensors-24-07216-f013:**
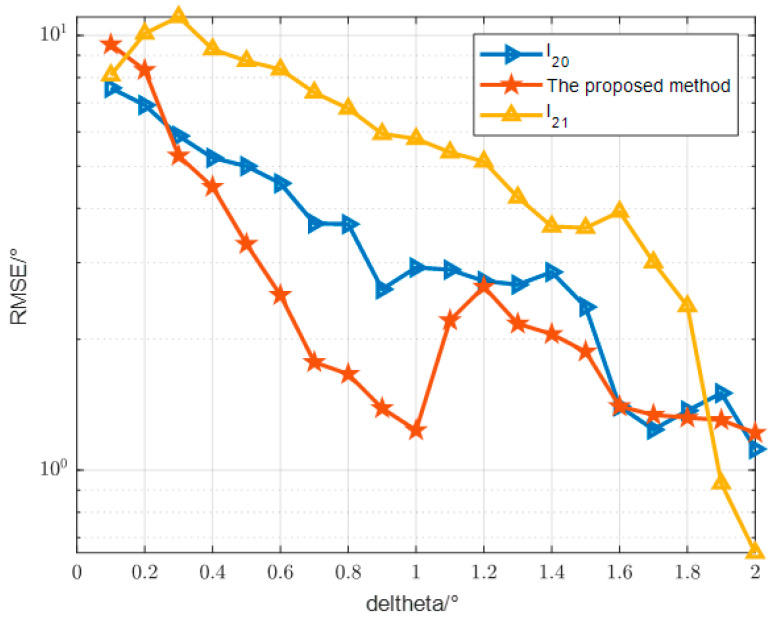
Relationship between direction finding error and angular interval.

**Figure 14 sensors-24-07216-f014:**
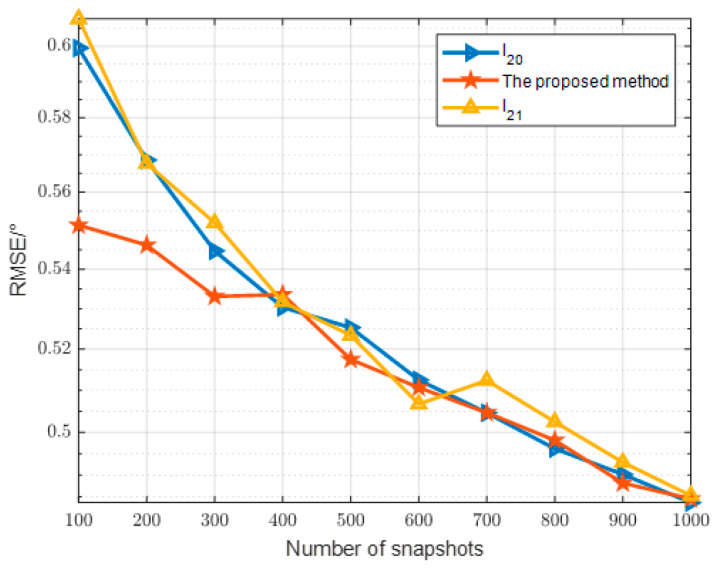
Algorithm performance versus number of snapshots.

**Figure 15 sensors-24-07216-f015:**
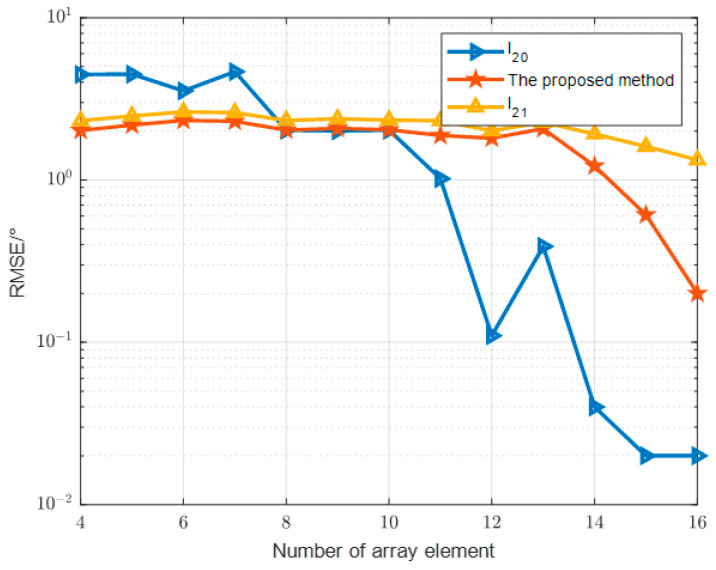
Algorithm performance versus number of array elements.

**Figure 16 sensors-24-07216-f016:**
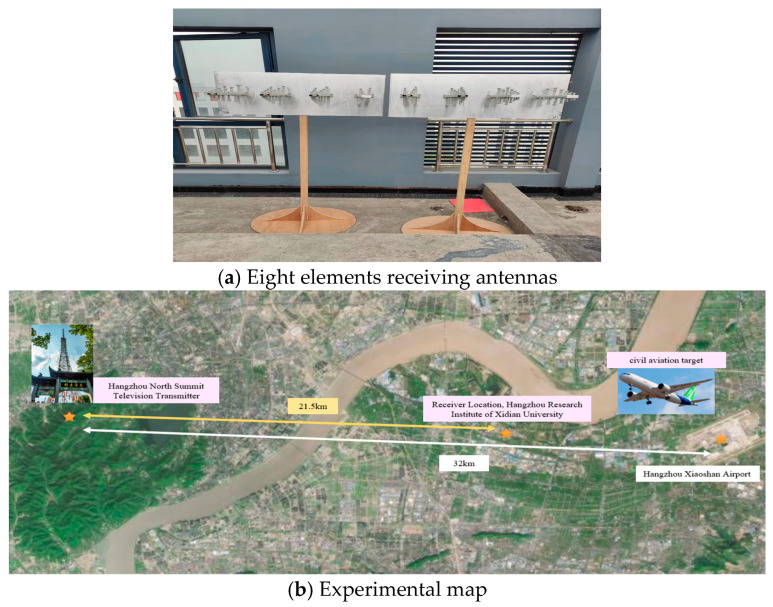
Antennas and map used in the experiment.

**Figure 17 sensors-24-07216-f017:**
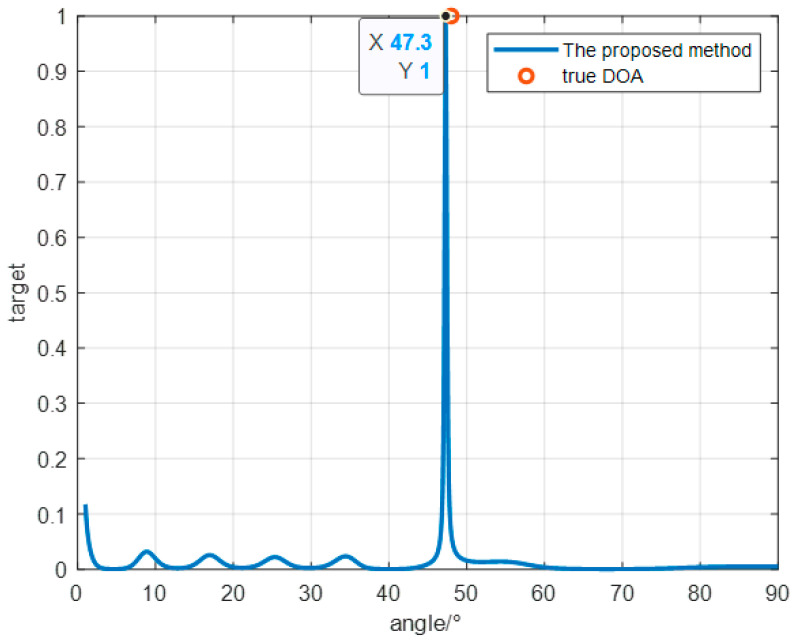
Experimental results of the measured data.

**Table 1 sensors-24-07216-t001:** Experiment settings.

**R**adiation **S**ource **L**ocation	(0, 0)
**R**eceiving **S**tation **L**ocation	(10, 0)
**T**arget **P**osition	(13.5, 50)
**N**umber of **A**rray **E**lements	64
**S**napshot **C**ount	1024
**S**patial **M**eshing	0.1°

## Data Availability

Due to some data privacy issues, the data used in this article cannot be provided.

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
