# Peer review of "A New Method for Joint Sparse DOA Estimation"

_sensors, 2024, doi:10.3390/s24227216_

Round 1
Reviewer 1 Report
Comments and Suggestions for Authors
This paper addresses the problem of Direction of Arrival (DOA) estimation in passive radar systems, focusing on the effective utilization of multi-snapshot data and its application in sparse reconstruction. The proposed model introduces a hyperbolic tangent function as a substitute for the non-convex mixed norm problem, transforming it into a solvable convex optimization issue. This research is valuable for enhancing the accuracy and spatial resolution of DOA estimation, especially under conditions with moderate signal-to-noise ratio (SNR) and close target angles.
1. The paper proposes a novel method for multi-snapshot data utilization, employing a hyperbolic tangent model to replace the traditional non-convex problem, showing some theoretical innovation. However, while the theoretical derivation is detailed, it lacks sufficient physical explanation and intuitive clarity. Moreover, the applicability and stability of the method in real-world scenarios have not been adequately validated through experiments, particularly under low SNR and complex environments. There is also a lack of discussion on potential algorithmic improvements or optimizations.
2. The structure of the paper is clear, with a logical flow from background introduction to problem modeling, method proposal, and validation. However, during the presentation of the model and algorithm derivation, the use of symbols and terminology is not sufficiently consistent, which might confuse the readers. Furthermore, the experimental design in the model validation section lacks detailed explanation of the comparison groups, making it less effective in demonstrating the superiority of the proposed method.
3. The experimental section shows the performance of DOA estimation under single-target, multi-target, and different numbers of array elements, comparing traditional methods with the proposed one. However, the SNR settings in the experiments are somewhat idealized, not considering the performance under low SNR and highly interfered environments. Moreover, the explanations for the figures are too brief, failing to detail the strengths, weaknesses, and performance differences of each algorithm under varying conditions.
4. The references cited include numerous classical and recent studies, demonstrating the authors' deep understanding of the field. However, some citations are either inaccurate or lack detailed descriptions. Especially for core theories and models, the citations are too brief, and there should be more in-depth discussions of related works. The authors should add the following references about DOA estimation:
[J1] X. T. Meng, B. X. Cao, F. G. Yan, M. Greco, F. Gini, Y. Zhang. Real-valued MUSIC for efficient direction of arrival estimation with arbitrary arrays: Mirror suppression and resolution improvement[J]. Signal Processing, 2023, 202: 108766.
Comments on the Quality of English LanguageThe English writing should be improved.
Author Response
I am very glad to receive such a professional review from you. Your comments are like a bright light that illuminates the blind spots in the project for us and its profound insight and forward-looking thinking undoubtedly point out the direction for our work.
Comments1: The paper proposes a novel method for multi-snapshot data utilization, employing a hyperbolic tangent model to replace the traditional non-convex problem, showing some theoretical innovation. However, while the theoretical derivation is detailed, it lacks sufficient physical explanation and intuitive clarity. Moreover, the applicability and stability of the method in real-world scenarios have not been adequately validated through experiments, particularly under low SNR and complex environments. There is also a lack of discussion on potential algorithmic improvements or optimizations.
Response1: We have updated the presentation in the modelling phase to make it more intuitive and accurate to read. We used radar echo data from a real scenario experiment in Hangzhou to verify the applicability and stability of the method in real scenarios, which is also shown in the paper. New simulation experiments in low SNR and complex environments are also included and discussed, as shown in Fig. 11. The applicability and optimization of the potential algorithms are discussed in a set of comparative experiments.
Comments 2: The structure of the paper is clear, with a logical flow from background introduction to problem modeling, method proposal, and validation. However, during the presentation of the model and algorithm derivation, the use of symbols and terminology is not sufficiently consistent, which might confuse the readers. Furthermore, the experimental design in the model validation section lacks detailed explanation of the comparison groups, making it less effective in demonstrating the superiority of the proposed method.
Response2: The issue of lack of consistency in the use of notation and terminology during the presentation of the model and algorithm derivation e.g. M and m denoting the total number of arrays and the mth arrays has been addressed. The issue of lack of detailed description of the comparison group in the experimental design of the comparison experiments has been enriched in the experimental analyses of the comparison group, as reflected in Figures 10 to 15.
Comments3: The experimental section shows the performance of DOA estimation under single-target, multi-target, and different numbers of array elements, comparing traditional methods with the proposed one. However, the SNR settings in the experiments are somewhat idealized, not considering the performance under low SNR and highly interfered environments. Moreover, the explanations for the figures are too brief, failing to detail the strengths, weaknesses, and performance differences of each algorithm under varying conditions.
Response3: An effect experiment close to 0dB SNR in a real scene is designed, and two new comparison experiments for the number of snapshots and the number of array elements are added to show the characteristics of the proposed method in multiple dimensions.
Comments 4: The references cited include numerous classical and recent studies, demonstrating the authors' deep understanding of the field. However, some citations are either inaccurate or lack detailed descriptions. Especially for core theories and models, the citations are too brief, and there should be more in-depth discussions of related works. The authors should add the following references about DOA estimation:
[J1] X. T. Meng, B. X. Cao, F. G. Yan, M. Greco, F. Gini, Y. Zhang. Real-valued MUSIC for efficient direction of arrival estimation with arbitrary arrays: Mirror suppression and resolution improvement[J]. Signal Processing, 2023, 202: 108766.
Response4: Your proposal to introduce more in-depth references at the core theory and model, we refer to this article in our signal modelling, the method is based on spatial spectrum estimation of DOA estimation, but its modelling is still worthy of reference.
Reviewer 2 Report
Comments and Suggestions for Authors
In this work, the authors proposed a new method for Direction of Arrival (DOA) estimation in passive radar systems by leveraging multi-snapshot data. The authors successfully demonstrated the novelty of their work through simulations. However, there are some concerns of this reviewer to be addressed. Please find below my comments.
- The authors claim their method improves DOA estimation accuracy, but the presented simulation results lack details on the real-world validation. Can the authors provide experimental evidence from actual radar data?
- The introduction of the hyperbolic tangent model is central to this paper. Can the authors provide more theoretical justification and derivations to explain why this model performs better than conventional non-convex methods in high-SNR scenarios?
- The paper emphasizes the use of multi-snapshot data for improved DOA estimation. How does the performance scale with increasing snapshots? Can the authors provide a complexity analysis for different snapshot quantities?
- The simulation setup seems idealized. What are the effects of real-world noise (e.g., correlated noise, multipath interference) on the algorithm’s performance? More detailed analysis is required.
- The presented RMSE plots do not cover higher-resolution grids. Also, the proposed algorithm for sparse recovery relies on specific assumptions about array geometry. How robust is the method to deviations in antenna spacing or phase errors? The authors are advised to discuss this in more details.
- The computational efficiency of the method under practical settings, particularly with large-scale antenna arrays, is unclear. Can the authors discuss how the algorithm’s performance scales with the number of array elements and how it can be optimized for real-time applications?
- The simulations assume ideal conditions for the number of array elements. How does the method perform with a reduced number of elements, especially under hardware constraints typical of passive radar systems? This analysis would greatly benefit the researchers.
The English language is generally good and clear, with only minor grammatical adjustments needed for better clarity.
Author Response
I am very glad to receive such a professional review from you. Your comments are like a bright light that illuminates the blind spots in the project for us and its profound insight and forward-looking thinking undoubtedly point out the direction for our work.
Comments 1: The authors claim their method improves DOA estimation accuracy, but the presented simulation results lack details on the real-world validation. Can the authors provide experimental evidence from actual radar data?
Response 1: We used radar echo data from a real scenario experiment conducted in Hangzhou to verify the applicability and stability of the method in real scenarios, the details of which are also reflected in the paper.
Comments 2: The introduction of the hyperbolic tangent model is central to this paper. Can the authors provide more theoretical justification and derivations to explain why this model performs better than conventional non-convex methods in high-SNR scenarios?
Response2: The improvement of the method proposed in this paper over traditional non-convex methods is due to the many mathematically optimizable properties described in the paper, such as conductivity, monotonicity, etc. This makes the effect of environmental changes on the model in practical applications smaller than that of a non-convex optimization.
Comments 3: The paper emphasizes the use of multi-snapshot data for improved DOA estimation. How does the performance scale with increasing snapshots? Can the authors provide a complexity analysis for different snapshot quantities?
Response3 The experiments in Figure 14 of the paper were supplemented and analyzed with measurement errors at different number of snapshots. The analysis of complexity is now not quantified.
Comments 4: The simulation setup seems idealized. What are the effects of real-world noise (e.g., correlated noise, multipath interference) on the algorithm’s performance? More detailed analysis is required.
Response 4: Your question fits well with the reality that the impact of real-world noise on algorithms is an issue worth investigating. However, resources are limited at present, and the modelling of such noise is itself a problem worth exploring deeply, and no simulation in this area has been carried out yet. However, we have conducted real-world data to validate the results.
Comments 5: The presented RMSE plots do not cover higher-resolution grids. Also, the proposed algorithm for sparse recovery relies on specific assumptions about array geometry. How robust is the method to deviations in antenna spacing or phase errors? The authors are advised to discuss this in more details.
Response5: We changed the axes of the comparison test to show them in logarithmic form, which helps in improving the visibility. We added the analysis of the phase error of the antenna array flow pattern to the experiment in Figure 11.
Comments 6: The computational efficiency of the method under practical settings, particularly with large-scale antenna arrays, is unclear. Can the authors discuss how the algorithm’s performance scales with the number of array elements and how it can be optimized for real-time applications?
Response 6: The newly added experiment in Fig. 15 explores the effect of antenna array elements on the performance of the three algorithms. How to optimize for real-time applications is a question worth exploring in depth, which we will discuss in the follow-up.
Comments 7: The simulations assume ideal conditions for the number of array elements. How does the method perform with a reduced number of elements, especially under hardware constraints typical of passive radar systems? This analysis would greatly benefit the researchers.
Response 7: In the experiment in Fig. 11 we design an experiment to approximate the signal-to-noise ratio as well as the number of array elements in a real scenario. The antenna array element number of a typical passive radar receiver module is usually set to 8, and there may be random perturbations in each channel of the antenna. We simulate and analyze this perturbation embodied in the phase.
Round 2
Reviewer 1 Report
Comments and Suggestions for Authors
I don't have the additional comments, where the authors have addressed all of my comments.
Reviewer 2 Report
Comments and Suggestions for Authors
Thank you for addressing he concerns of this reviewer. I have no more comments.
Comments on the Quality of English LanguageAcceptable